# First Report on Several NO-Donor Sets and Bidentate Schiff Base and Its Metal Complexes: Characterization and Antimicrobial Investigation

**Amira A. Mohamed** [1] , **Abeer A. Nassr** [2], **Sadeek A. Sadeek** [2,*], **Nihad G. Rashid** [3] **and Sherif M. Abd El-Hamid** [4]

1   Department of Basic Science, Zagazig Higher Institute of Engineering and Technology, Zagazig 44519, Egypt; aa.adaim@science.zu.edu.eg
2   Department of Chemistry, Faculty of Science, Zagazig University, Zagazig 44519, Egypt; abeernassr@gmail.com
3   Ministry of Education, Babylon 51001, Iraq; rashidnihad8@gmail.com
4   Department of Basic Science, Higher Future Institute of Engineering and Technology, Mansoura 35516, Egypt; murap5@mans.edu.eg or sherifmohamed266@gmail.com
*   Correspondence: s_sadeek@zu.edu.eg; Tel.: +20-01220057510; Fax: +20-0553208213

**Abstract:** The condensation product of the reaction between aniline and salicylaldehyde was a 2-(2-hydroxybenzylidinemine)—aniline Schiff base bidentate ligand (**L**). **L** was used to generate complexes by interacting with the metal ions lanthanum(III), zirconium(IV), yttrium(III), and copper(II), in addition to cobalt(II). Various physicochemical techniques were utilized to analyze the synthesized **L** and its metal chelates, including elemental analysis (CHN), conductimetry ($\Lambda$), magnetic susceptibility investigations ($\mu_{eff}$), Fourier-transform infrared spectroscopy (FT-IR), proton nuclear magnetic resonance ($^1$H NMR) spectroscopy, ultraviolet–visible (UV-Vis.) spectrophotometry, and thermal studies (TG/DTG). FT-IR revealed that the **L** molecule acted as a bidentate ligand by binding to metal ions via both the oxygen atom of the phenolic group in addition to the nitrogen atom of the azomethine group. Additionally, $^1$H NMR data indicated the formation of complexes via the oxygen atom of the phenolic group. An octahedral geometrical structure for all of the chelates was proposed according to the UV-Vis. spectra and magnetic moment investigations. Thermal analysis provided insight into the pattern of **L** in addition to its chelates' breakdown. In addition, the investigation furnished details on the chelates' potential chemical formulas, the characteristics of adsorbed or lattice $H_2O$ molecules, and the water that is coordinated but separated from the structure at temperatures exceeding 120 °C. The thermodynamic parameters utilizing Coats–Redfern in addition to Horowitz–Metzger equations were studied. The antimicrobial effectiveness of **L** and its chelates against distinct species of bacteria and fungi was studied using the disc diffusion method. Cu(II) and Y(III) chelates had significant antimicrobial activity against *Staphylococcus aureus* and *Micrococcus luteus*.

**Keywords:** Schiff base; metal complexes; spectroscopy; TG/DTG; antimicrobial effectiveness

## 1. Introduction

Recent developments in bioinorganic chemistry and medicine have improved coordination chemistry to research transition metal complexes of Schiff bases [1]. Schiff bases with the general formula $R_2C = NR$ produced via the condensation of an amine with a ketone or aldehyde are well known and can be utilized as starting materials to create a variety of chemical compounds for commercial and biological applications [2,3]. Schiff bases can be utilized in dyes, pigments, catalysts, and polymer stabilizers, and can serve as antioxidants against reactive oxygen species [4–6]. In addition, Schiff bases are employed as ligands to create coordination compounds when coupled with metal ions [7,8]. Many biological applications of Schiff bases and their chelates are possible, due to their antibacterial, anticancer, antifungal, antimalarial, and antiviral properties [9–11].

The unique characteristics of Schiff bases and their chelates include stability, low toxicity, aromaticity, redox activity, lipophilicity, various membrane penetration characteristics, and abnormal metabolism [12–15]. Aniline and salicylaldehyde were condensed to produce the appropriate Schiff base, 2-(2-hydroxybenzylidinemine)-aniline (**L**) (Scheme 1), which is categorized as a liquid crystal with structural units that are rod- or disc-shaped [16–20] and possesses mesomorphism; it has been extensively researched and utilized in various scientific, technological, and industrial fields. Additionally, **L** possesses photochromic qualities [15–22]. N-salicylidene anilines can form chelates with certain metal ions such as Ni, Rh, Cu, Pd, Co, and Pt. The geometric structure of the molecules is modified by these complexes, leading to sematic mesomorphism [23–26]. Through extensive literature research, it has been discovered that there are no reports of work being conducted on the 2-(2-hydroxybenzylidinemine)-aniline (**L**) Schiff base with the metal ions lanthanum(III), zirconium(IV), yttrium(III), and copper(II), in addition to cobalt(II). The aim of this study was to synthesize **L** metal complexes; the purpose of selecting the **L** ligand was to investigate how the efficacy of the biological properties of **L** would be affected by altering the atomic volume, atomic weight, and oxidation state of certain important elements using constant counter ions that participate in various functions and structures in biological systems, which could lead to an improvement in the effects or stability.

(**L**)

**Scheme 1.** Structure of 2-(2-hydroxybenzylidinemine)-aniline (**L**).

The resulting mononuclear metal complexes were characterized using spectroscopic methods like UV-Vis., $^1$H NMR, FT-IR, CHN, $\Lambda$, and $\mu_{eff}$ in addition to TG and DTG. The antimicrobial activity of **L** and its chelates toward bacteria and fungi was evaluated through in vitro testing, and the resulting data were statistically analyzed.

## 2. Materials and Methods

### 2.1. Materials and Reagents

Only chemicals and solvents that met the criteria for analytical reagent grade and had the highest level of purity were utilized. Lanthanum chloride heptahydrate, cobalt chloride hexahydrate, yttrium chloride hexahydrate, copper chloride dihydrate, zirconium chloride octahydrate, ferric chloride hexahydrate, salicylaldehyde, potassium chromate, aniline, dimethylformamide (DMF), ethanol absolute, dimethyl sulfoxide (DMSO), and silver nitrate were purchased from Fluka and Sigma-Aldrich Chemical Co (State Louis, MI, USA). Each glass was submerged in a chromatic solution ($K_2Cr_2O_7$ + conc.$H_2SO_4$) and was thereafter cleaned with bidistilled water and placed in an oven for drying at one hundred degrees Celsius.

### 2.2. Preparation of 2-(2-Hydroxybenzylidinemine)-aniline (L)

Twenty mmol of aniline (1.81 mL) was added to twenty mmol of salicylaldehyde (2.12 mL) in fifty mL ethanol, and the mixture was refluxed for eight hours with two mL of glacial acetic acid (Scheme 2). The mixture was concentrated until it reached a volume of eight mL using a water bath and was subsequently cooled down to zero degrees Celsius. After filtering, the light yellow solid (**L**) was dried over anhydrous $CaCl_2$.

**Scheme 2.** Preparation of 2-(2-hydroxybenzylidinemine)-aniline (**L**).

### 2.3. Synthesis of *L* Metal Complexes

The gray solid complex [Co(**L**)(H$_2$O)$_2$]Cl$_2$·4H$_2$O (**1**) was created by reacting 2 mmol (0.394 g) of **L** with 1 mmol (0.237 g) of CoCl$_2$·6H$_2$O in 40 mL ethanol which was refluxed for 6 h. After slow evaporation, a gray precipitate was produced and dried under vacuum over anhydrous CaCl$_2$. Dark brown [Cu(**L**)(H$_2$O)$_2$]Cl$_2$·3H$_2$O (**2**), dark green [Y(**L**)(H$_2$O)$_2$]Cl$_3$ (**3**), green [ZrO(**L**)(H$_2$O)]Cl$_2$·4H$_2$O (**4**) and Pale green [La(**L**)(H$_2$O)$_2$]Cl$_3$·4H$_2$O (**5**) were prepared using a method comparable to the one mentioned earlier above utilizing CoCl$_2$·6H$_2$O, YCl$_3$·6H$_2$O, ZrOCl$_2$·8H$_2$O in addition to LaCl$_3$·7H$_2$O, respectively, in ethanol as the solvent in a 2:1 (**L**:M) molar ratio.

### 2.4. Instruments

Fourier-transform infrared spectra were captured in KBr discs by utilizing an FT-IR 460 PLUS Spectrophotometer (International Equipment Trading Ltd., Mundelein, IL, USA), with a wavenumber range spanning from 4000 to 400 cm$^{-1}$. Using DMSO-d$_6$ as the solvent, the $^1$H NMR spectra were obtained using a Varian Mercury VX-300 NMR Spectrometer (International Equipment Trading Ltd., Mundelein, IL, USA). The compounds were dissolved in DMSO and used to capture electronic absorption spectra using a Shimadzu UV-3101PC (International Equipment Trading Ltd., Mundelein, IL, USA). A TGA-50H (International Equipment Trading Ltd., Mundelein, IL, USA) instrument was employed for performing the TG-DTG measurements in an N$_2$ environment (International Equipment Trading Ltd., Mundelein, IL, USA), with the specimen's mass being precisely measured in an aluminum crucible. The measurements were conducted at temperatures spanning from room temperature to 1000 °C. Three analytical techniques were used for estimating the percentage of metal content: thermogravimetry, atomic absorption, and complexometric titration. Atomic absorption analysis was performed using a PYE-UNICAM SP 1900 spectrometer (International Equipment Trading Ltd., Mundelein, IL, USA) set up using the proper lamp [27,28]. A Perkin Elmer model 2400 CHN elemental analyzer (Hunan sundy Science and Technology Co., Ltd., Changsha, China) was utilized to conduct the studies. The melting points were determined using a Buchi device. The magnetic susceptibilities of the powdered materials were examined at room temperature using a Sherwood Scientific magnetic scale (Hunan sundy Science and Technology Co., Ltd., Changsha, China) and Hg[Co(CSN)$_4$] as a calibrant, measured using a Gouy balance (Hunan sundy Science and Technology Co., Ltd., Changsha, China). The molar conductance of the ligand and its chelates in DMF at a concentration of $1 \times 10^{-3}$ M was investigated using CONSORT K410 (Hunan sundy Science and Technology Co., Ltd., Changsha, China). Each test was conducted with freshly prepared solutions at room temperature.

### 2.5. Antimicrobial Investigation

The antimicrobial efficiency was evaluated against five common microbial strains collected from the Egyptian Pharmaceutical Industries Corporation (EPICO), Egypt. These pathogens included Gram-positive bacteria (*Staphylococcus aureus* ATCC 6538 and *Micrococcus luteus* TCC 10240), Gram-negative bacteria (*Escherichia coli* ATCC 10536 and *Salmonella typhi* ATCC 14028) and one fungal species (*Candida albicans* ATCC 10231). The Kirby Bauer disc diffusion approach, which adheres to the guidelines established by the Clinical Laboratory and Standards Institute (CLSI, 2012), was utilized to evaluate the antimicrobial efficacy

of the examined compounds [29,30]. Microbial samples were added to Mueller–Hinton broth and incubated at thirty-five degrees Celsius until the degree of turbidity reached 0.5 McFarland standard or greater. In order to achieve the turbidity level equivalent to the 0.5 McFarland standards, the suspension's turbidity was adjusted using sterile saline. The aseptic pouring technique was used to pour sterile Mueller–Hinton agar plates. Within fifteen minutes of adjusting the turbidity of the inoculum suspension, a sterile cotton swab was immersed in the modified suspension. The swab was rotated several times above the liquid level and firm pressure was applied against the inner wall of the tube. By doing so, any excess inoculum on the swab will be eliminated. The cotton swab was swiped across the dried Müeller–Hinton agar plate to inoculate the plate. To achieve a uniform distribution of the inoculum, the plate was swiped twice more while turning it at approximately 60-degree intervals each time. As a final step, the edge of the agar was wiped with a swab. Before placing the drug-impregnated discs, the lid was left slightly open for a period of three to five minutes, but not exceeding 15 min, to enable evaporation of any excess surface moisture. To achieve full interaction with the agar surface, small and circular filter paper discs with a diameter of approximately five millimeters that had been sterilized and preloaded with eight liters of the test specimens were positioned on the plates. After being inverted and incubated at a temperature of 35–37 °C for a duration of 16–20 h, the diameter of the zones of inhibition was measured [31,32]. The degree of microbial growth inhibition was determined relative to that of the positive control. Using the following equation, the activity index % of the compounds was determined.

$$\% \text{ Activity Index} = \frac{\text{Zone of inhibition by test compound (diameter)}}{\text{Zone of inhibition by standard (diameter)}} \times 100 \quad (1)$$

## 3. Results and Discussion

### 3.1. Elemental Composition and Molar Conductance

Table 1 shows the data from the elemental analyses, molecular formulas, melting temperatures, yields of the compounds, and molar conductance values. The calculated values and the findings of the elemental analysis for **L** and its complexes were in excellent accordance and revealed that the stoichiometry of our complexes was 2:1 (**L**:metal). The complexes are solid hydrates with varying degrees of hydration, air-stable at normal temperature with high melting points, soluble in DMF in addition to DMSO, but insoluble in most common organic solvents. Molar conductance values of the complexes were in the range of 92.45–125.00 $\Omega$ cm$^2$ mol$^{-1}$ (Table 1). The values adequately confirmed the electrolytic nature of the chelates and agree with the qualitative analysis test that demonstrated that chloride ions exist outside the coordination sphere [9,33,34].

**Table 1.** Elemental composition and analytical data for **L** and its chelates.

| Compounds MW (M.F.) | Yield% | M.P./°C | Color | (Calc.) Found (%) | | | | | Λ (S cm$^2$ mol$^{-1}$) |
|---|---|---|---|---|---|---|---|---|---|
| | | | | C | H | N | Cl | M | |
| **L** 197 (C$_{13}$H$_{11}$NO) | - | 50 | Light-yellow | (79.18) 79.01 | (5.58) 5.52 | (7.10) 6.89 | - | - | 1.45 |
| **(1)** 631.83 (CoC$_{26}$H$_{34}$N$_2$O$_8$Cl$_2$) | 88.34 | 215 | Gray | (49.38) 49.20 | (5.38) 5.23 | (4.43) 4.26 | (11.22) 11.13 | (9.32) 9.21 | 95.84 |
| **(2)** 618.44 (CuC$_{26}$H$_{32}$N$_2$O$_7$Cl$_2$) | 80.69 | 210 | Dark-brown | (50.44) 50.28 | (5.17) 5.10 | (4.52) 4.41 | (11.46) 11.32 | (10.27) 10.18 | 92.45 |
| **(3)** 622.25 (YC$_{26}$H$_{26}$N$_2$O$_4$Cl$_3$) | 90.25 | 220 | Dark-green | (50.14) 49.88 | (4.17) 4.10 | (4.49) 4.38 | (17.09) 17.00 | (14.28) 14.12 | 125.00 |
| **(4)** 662.12 (ZrC$_{26}$H$_{32}$N$_2$O$_8$Cl$_2$) | 82.24 | >300 | Green | (47.12) 46.89 | (4.83) 4.97 | (4.22) 4.18 | (10.70) 10.62 | (13.77) 13.65 | 96.10 |
| **(5)** 737.25 (LaC$_{26}$H$_{34}$N$_2$O$_8$Cl$_3$) | 85.75 | >300 | Pale-green | (42.31) 42.17 | (4.61) 4.51 | (3.79) 3.70 | (14.42) 14.31 | (18.84) 18.70 | 123.80 |

### 3.2. FT-IR Spectra and Mode of Bonding

In order to facilitate the assignment of the observed bands in the free **L** and its chelates and to identify the coordination sites that may be involved in complex formation, a detailed examination of the FT-IR spectra of **L** and its chelates was accomplished. The principal stretching frequencies of the FT-IR spectra for **L** and its chelates are illustrated in Table 2 and Figure S1. It was determined that the wide and moderately intense bands (shown in Figure S1) with wavenumbers ranging from 3396 to 3473 cm$^{-1}$ corresponded to the stretching vibrations of v(O-H), which could be attributed to either the lattice, coordinated water, or phenolic group [35,36]. Our focus was first directed toward the vibration bands of the phenolic and azomethine groups, which were thought to be involved in the interaction of the **L** compound with the metal cations. A vibration band occurring at 1612 cm$^{-1}$ was credited to the stretching vibration of the C=N bond in the azomethine group of the **L** compound. However, this band shifted to lower wavenumbers in the chelate spectra around 1601 cm$^{-1}$; this demonstrated the azomethine nitrogen involvement in the coordination [37–41]. The v(Zr=O) in complex (4) was found as a medium band at 832 cm$^{-1}$ [42]. New bands were present in the spectra of the synthesized chelates with diversified intensities at 663 and 570 cm$^{-1}$ for (1); 687 and 575 cm$^{-1}$ for (2); 686 and 580 cm$^{-1}$ for (3); 674 and 555 cm$^{-1}$ for (4); and 686 and 565 cm$^{-1}$ for complex (5), which were missing in the spectrum of **L** and indicates the complexation between **L** as bidentate and metal ions [43–46]. The estimated chemical structures for our chelates are shown in Scheme 3 in accordance with the information that was obtained.

**Scheme 3.** The coordination mode of **L** with Co(II), Cu(II), Zr(IV), Y(III), and La(III) metal ions. M = Co(II) and Cu(II) for *n* = 2 and M = La(III) and Y(III) for *n* = 3.

**Table 2.** Infrared wavenumber (cm$^{-1}$) of **L** and its metal complexes.

| L | (1) | (2) | (3) | (4) | (5) | Assignments |
|---|---|---|---|---|---|---|
| 3426 mbr | 3408 mbr | 3473 m | 3413 mbr | 3396 mbr | 3398 mbr | v(O–H); H$_2$O |
| 1612 vs | 1600 vs | 1603 vs | 1601 vs | 1600 vs | 1602 vs | v(C=N) pyridine ring |
|  |  |  |  | 832 m |  | v(Zr=O) |
|  | 688 w | 687 w | 686 w | 674 vw | 686 w | v(M–O) and |
|  | 570 w | 575 w | 580 w | 555 w | 565 w | v(M–N) |

Keys: s = strong, w = weak, v = very, m = medium, br = broad, v = stretching.

### 3.3. Electronic Absorption Studies and Magnetic Moment Measurements

The UV-Vis. spectra for the compounds were detected over a range of wavelengths from 200 to 800 nm (Figure S2) and their electronic spectra data (Table 3) demonstrate the molecular geometry of the complexes. The ligand **L** displayed three absorption bands which were assigned to the intra-molecular transitions (π–π*) at 270 nm (37,037 cm$^{-3}$), 318 nm (31,446 cm$^{-1}$), and 340 nm (29,411 cm$^{-1}$) revealed to be (n–π*) transitions [40]. The shift of intramolecular transitions for **L** to slightly higher or lower wavelengths in the chelates is attributed to the complexation between **L** and metal ions [45,46]. New bands in

the spectra of chelates found in the range 415–470 nm (24,096–21,276 cm$^{-7}$) demonstrate the ligand metal charge transfer [9,10]. An absorption band occurring at 17,241 cm$^{-1}$ in the electronic spectrum of the Co(II) complex which possesses a magnetic moment value of 5.20 B.M.; this implies an octahedral geometry of the $4T_{1g}$ (F) $\rightarrow$ $4T_{1g}$ (P) transition with a 10 Dq value of 206 kJ/mole and CFSE of 206 + 2p [47,48]. A band occurring at 16,806 cm$^{-1}$ was detected in the Cu(II) complex, with a magnetic moment value of 1.70 B.M., which could be attributed to the $2B_{1g}$ $\rightarrow$ $2E_{1g}$ transition with a 10 Dq value of 201 kJ/mole and CFSE of 201 + 4p corresponding to an octahedral geometry [49–51]. Since lanthanum(III), zirconium(IV), and yttrium(III) chelates were diamagnetic as predicted from their electronic configuration (d$^{10}$), no d-d transitions were observed in the spectra of these chelates. The molar absorptivity ($\varepsilon$) of complexes specified from their electronic spectra was documented (Table 3) using the relationship A = $\varepsilon$cl (2), where A = absorbance, c = 1.0 $\times$ 10$^{-3}$ M, and l = length of the cell (1 cm).

**Table 3.** Ultraviolet–visible spectroscopy of L and its chelates.

| Compounds | Peak | | Assignment | $E$ $(M^{-1}cm^{-1}) \times 10^4$ | 10Dq | | C′FSE | $\mu_{eff}$ (B.M) |
| | nm | cm$^{-1}$ | | | cm$^{-1}$ | kJ/mol | | |
|---|---|---|---|---|---|---|---|---|
| **L** | 270 | 37,037 | $\pi \rightarrow \pi^*$ | 0.600 | | | | |
| | 318 | 31,446 | $n \rightarrow \pi^*$ | 0.723 | | | | |
| | 340 | 29,411 | $n \rightarrow \pi^*$ | 0.769 | | | | |
| **(1)** | 275 | 36,363 | $\pi \rightarrow \pi^*$ | 2.150 | | | | 5.20 |
| | 312 | 32,051 | $n \rightarrow \pi^*$ | 1.567 | | | | |
| | 345 | 28,958 | $n \rightarrow \pi^*$ | 1.425 | 17,241 | 206 | 206 + 2p | |
| | 450 | 22,222 | CT | 0.825 | | | | |
| | 580 | 17,241 | $4T_{1g}$ (F) $\rightarrow$ $4T_{1g}$ (P) | 0.450 | | | | |
| **(2)** | 287 | 41,666 | $\pi \rightarrow \pi^*$ | 1.400 | | | | 1.70 |
| | 365 | 27,397 | $n \rightarrow \pi^*$ | 0.900 | | | | |
| | 415 | 24,096 | CT | 0.443 | 16,806 | 201 | 201 + 4p | |
| | 595 | 16,806 | $2B_{1g} \rightarrow 2E_{1g}$ | 0.350 | | | | |
| **(3)** | 270 | 37,037 | $\pi \rightarrow \pi^*$ | 2.150 | | | | |
| | 325 | 30,769 | $n \rightarrow \pi^*$ | 2.100 | | | | |
| | 338 | 29,585 | $n \rightarrow \pi^*$ | 2.000 | | | | |
| | 440 | 22,727 | CT | 0.923 | | | | |
| **(4)** | 266 | 37,593 | $\pi \rightarrow \pi^*$ | 0.867 | | | | |
| | 348 | 28,735 | $n \rightarrow \pi^*$ | 0.426 | | | | |
| | 465 | 21,505 | CT | 0.328 | | | | |
| **(5)** | 270 | 37,037 | $\pi \rightarrow \pi^*$ | 0.630 | | | | |
| | 320 | 31,250 | $n \rightarrow \pi^*$ | 0.500 | | | | |
| | 337 | 29,673 | $N \rightarrow \pi^*$ | 0.490 | | | | |
| | 470 | 21,276 | CT | 0.375 | | | | |

### 3.4. Nuclear Magnetic Resonance

[1]H NMR spectra were obtained to confirm the hypothesized structure of the isolated chelates. Tetramethylsilane (TMS) was used as an internal standard to record the [1]H NMR spectra of **L** and its chelates. The results are outlined in Table 4 in addition to being displayed in Figure S3. The [1]H NMR spectrum of **L** revealed at a chemical shift of 8.95 ppm for the –N=CH azomethine group, at 6.96–7.94 ppm for the –CH aromatic group and the signal observed at a chemical shift of 13.09 ppm can be credited to the presence of the phenolic hydroxyl group [52–54]. Based on the observation of the proton (OH) for the phenolic group in the spectra of the chelates and its chemical shift, it is suggested that **L** interacts via the oxygen atom of the proton in the phenolic group. All signals of the free **L** were present in the complexes' spectra with some changes which demonstrated that **L** reacted with metal ions and formed metal complexes [55,56]. The proton signal, which can be linked to the existence of $H_2O$ molecules and present between 3.32 and 3.48 ppm, was consistent with the formula for chelates that have been proposed.

**Table 4.** [1]H NMR values (ppm) and tentative assignments for **L** and its metal complexes.

| L | (1) | (2) | (3) | (4) | (5) | Assignments |
|---|---|---|---|---|---|---|
| 2.48 | 2.40 | 2.52 | 2.50 | 1.82–2.70 | 2.51 | $\delta$H, –CH aliphatic (DMSO) |
| - | 3.48 | 3.37 | 3.32 | 3.40 | 3.33 | $\delta$H, $H_2O$ |
| 6.96–7.94 | 7.23–7.88 | 7.04–7.91 | 7.00–7.91 | 7.20–7.98 | 7.00–7.90 | $\delta$H, –CH aromatic |
| 8.95 | 8.93 | 8.92 | 8.96 | 8.96 | 8.96 | $\delta$H, –H–C=N |
| 13.09 | 13.00 | 13.02 | 13.01 | 13.20 | 13.00 | $\delta$H, –OH |

### 3.5. Thermal Analysis Studies (TG and DTG)

Thermogravimetric analyses (TG and DTG) of **L** and its chelates were utilized to study the thermal stabilization of these new chelates, determine whether $H_2O$ molecules are internal (coordinated) or external (crystalline) to the internal coordination sphere of the central metal ion, and eventually to discuss a comprehensive scheme for thermal decay of these compounds; the data are presented in Table 5 and Figure S4. The TG of **L** proceeded through one phase at $T_{max}$ (200 °C), resulting in $2C_4H_2 + H_2O + 0.5N_2 + 2.5C_2H_2$, exhibiting a mass loss of approximately 99.68% (calculated to be 100%). Complexes (**1**) and (**5**) decomposed in three phases and exhibited essentially comparable thermal behaviors. The initial phase is linked to the expulsion of water molecules from the crystal lattice, which happens at a $T_{max}$ of 57 and 95 °C. The next decomposition phase at a $T_{max}$ of 129 and 150 °C with loss of $2H_2O$ and $6C_4H_2 + 2H_2O$, respectively. In the final phase, the complexes undergo decay at two different $T_{max}$ values, 252, 597 and 265, 597 °C, resulting in a total mass loss of 84.50 (calc. 84.35) and 81.28 (calc. 81.16), respectively. The final residues after decay were CoO and 2C, as well as La. TG analysis of complex (**3**) started at 215 to 600 °C with a mass loss of 85.83% (calc. 85.72%) corresponding to a loss of $6C_4H_2 + 3HCl + 2H_2O + 2NH_3 + 2CO + 0.5H_2$. After complete thermolysis, the calculated and observed percentages of the residue were coincident with metallic yttrium. According to the TG curve, complexes (**2**) and (**4**) decomposed in two steps. The first step of degradation involves the loss of lattice water ($2H_2O$ and $4H_2O$) with a mass loss of 8.68 % (calc. 8.73%) and 10.84% (calc. 10.87%), respectively. The second step corresponds to loss of $11C_2H_2 + C_2N_2 + H_2O + 2CO + 2HCl_2$ and $6C_4H_2 + 2HCl + 2H_2O + C_2N_2 + 3H_2$ with a $T_{max}$ of 275, 402 and 604 °C leaving $CuO + 2C$ and $ZrO_2$ as final residues. The thermal residues were differentiated by infrared spectra, which showed the absence of all peaks characteristic to chelated L and instead the characteristic peaks for $CuO$ and $ZrO_2$ were found [57,58]. We obtained IR spectra for $CuO$ and $ZrO_2$ as shown in Figure S5 and XRD for $ZrO_2$ (Figure S6) was performed [59].

**Table 5.** $T_{max}$ (C) and mass loss (%) values of the decay phases for **L** and its chelates.

| Compounds | Decay Steps | $T_{max}$ (°C) | Mass Loss (%) | | Lost Species |
|---|---|---|---|---|---|
| | | | Calc. | Found | |
| **L** | Step one | 200 | 100.00 | 99.68 | $2C_4H_2 + H_2O + 0.5N_2 + 2.5C_2H_2$ |
| | Total loss | | 100.00 | 99.68 | |
| **(1)** | Step one | 57 | 11.39 | 11.38 | $4H_2O$ |
| | Step two | 129 | 5.69 | 5.68 | $2H_2O$ |
| | Step three | 252,597 | 67.27 | 67.44 | $6C_4H_2 + 2HCl + H_2O + 2NH_3$ |
| | Total loss | | 84.35 | 84.50 | |
| | Residue | | 15.65 | 15.50 | $CoO + 2C$ |
| **(2)** | Step one | 75,114 | 8.73 | 8.68 | $3H_2O$ |
| | Step two | 275,402 | 74.53 | 74.63 | $11C_2H_2 + C_2N_2 + H_2O + 2CO + 2HCl_2$ |
| | Total loss | | 83.26 | 83.31 | |
| | Residue | | 16.74 | 16.69 | $CuO + 2C$ |
| **(3)** | Step one | 227 | 85.72 | 85.83 | $6C_4H_2 + 3HCl + 2H_2O + 2NH_3 + 2CO + 0.5H_2$ |
| | Total loss | | 85.72 | 85.83 | |
| | Residue | | 14.28 | 14.17 | Y |

**Table 5.** *Cont.*

| Compounds | Decay Steps | $T_{max}$ (°C) | Mass Loss (%) | | Lost Species |
| | | | Calc. | Found | |
|---|---|---|---|---|---|
| **(4)** | Step one | 109 | 10.87 | 10.84 | $4H_2O$ |
| | Step two | 604 | 70.52 | 70.65 | $6C_4H_2 + 2HCl + 2H_2O + C_2N_2 + 3H_2$ |
| | Total loss | | 81.39 | 81.49 | |
| | Residue | | 18.61 | 18.51 | $ZrO_2$ |
| **(5)** | Step one | 95 | 9.76 | 9.73 | $4H_2O$ |
| | Step two | 150 | 45.57 | 45.53 | $6C_4H_2 + 2H_2O$ |
| | Step three | 265,597 | 25.83 | 26.02 | $3HCl + 2CO + 2NH_3 + 0.5H_2$ |
| | Total loss | | 81.16 | 81.28 | |
| | Residue | | 18.84 | 18.72 | La |

### 3.6. Calculation of Thermodynamic Parameters

The Coats–Redfern [60] in addition to Horowitz–Metzger [61] designs were applied to dynamically predetermine the thermodynamic dimensions of activation energy (Ea), enthalpy ($\Delta H^*$), free energy ($\Delta G^*$), and entropy ($\Delta S^*$) (Figure S7).

Coats–Redfern equations

$$\ln\ X = \ln\left[\frac{1-(1-\alpha)^{1-n}}{T^2(1-n)}\right] = \frac{-E^*}{RT} + \ln\left[\frac{AR}{\varphi E^*}\right] \text{for n} \neq 1 \tag{2}$$

where (n = 0, 0.33, 0.5, and 0.66).

$$\ln\ X = \ln\left[\frac{-\ln(1-\alpha)}{T^2}\right] = \frac{-E^*}{RT} + \ln\left[\frac{AR}{\varphi E^*}\right] \text{for n} = 1 \tag{3}$$

Horowitz–Metzger equations

$$\ln\ X = \ln\left[\frac{1-(1-\alpha)^{1-n}}{(1-n)}\right] = \frac{-E^*}{RT} + \ln\left[\frac{AR}{\varphi E^*}\right] \text{for n} \neq 1 \tag{4}$$

where (n = 0, 0.33, 0.5, and 0.66).

$$\ln\ X = \ln[-\ln(1-\alpha)] = \frac{-E^*}{RT} + \ln\left[\frac{AR}{\varphi E^*}\right] \text{for n} = 1 \tag{5}$$

$$\Delta H^* = E - RT \tag{6}$$

$$\Delta S^* = R\ \ln(Ah/k_B T_s) \tag{7}$$

$$\Delta G^* = \Delta H^* - T\ \Delta S^* \tag{8}$$

The higher $E_a$ values listed in Table 6 illustrate the thermo-stability of the complexes [62,63]. The elimination efficiency of chelated **L** will be slower than the preceding **L** due to the increasing $\Delta G^*$ value in the subsequent phases of breakdown, which will cause an accumulation from one phase to the next. In contrast to the earlier complex that consumes more energy, $T\Delta S^*$, this can be due to the structural rigidity of the residual complex following exclusion of one or more **L** through rearrangement before undergoing any modulation. The $\Delta S^*$ for all chelates had been set to be minus, which further defined their stability [64]. The positive values in $\Delta H^*$ denote an endothermic degradation process.

**Table 6.** Kinetic parameters for **L** and its chelates.

| Compounds | Decay Range (K) | $T_s$ (K) | Method | Parameters | | | | | R [a] | SD [b] |
| | | | | $E_a$ (kJ/mol) | A ($s^{-1}$) | $\Delta S^*$ (kJ/mol.K) | $\Delta H^*$ (kJ/mol) | $\Delta G^*$ (kJ/mol) | | |
|---|---|---|---|---|---|---|---|---|---|---|
| **L** | 315–917 | 473 | CR | 27.46 | 8.1567 | −0.231 | 23.57 | 111.75 | 0.986 | 0.163 |
| | | | HM | 43.95 | $1.43 \times 10^1$ | −0.226 | 40.02 | 147.20 | 0.970 | 0.238 |
| **(1)** | 298–393 | 330 | CR | 33.89 | $1.26 \times 10^2$ | −0.205 | 31.15 | 98.96 | 0.982 | 0.141 |
| | | | HM | 36.36 | $3.81 \times 10^3$ | −0.177 | 33.62 | 92.10 | 0.974 | 0.168 |
| **(2)** | 463–603 | 548 | CR | 53.45 | $4.59 \times 10^2$ | −0.199 | 48.91 | 157.96 | 0.978 | 0.190 |
| | | | HM | 63.24 | $4.48 \times 10^3$ | −0.180 | 58.68 | 157.35 | 0.966 | 0.236 |
| **(3)** | 423–610 | 500 | CR | 38.86 | $3.19 \times 10^1$ | −0.220 | 34.71 | 144.90 | 0.979 | 0.190 |
| | | | HM | 46.26 | $2.51 \times 10^2$ | −0.203 | 42.11 | 143.73 | 0.963 | 0.250 |
| **(4)** | 355–435 | 382 | CR | 32.39 | $1.31 \times 10^2$ | −0.206 | 29.21 | 108.07 | 0.982 | 0.150 |
| | | | HM | 45.15 | 9.240 | −0.228 | 41.94 | 129.25 | 0.972 | 0.184 |
| **(5)** | 608–463 | 538 | CR | 71.00 | $1.04 \times 10^2$ | −0.211 | 66.53 | 180.13 | 0.981 | 0.157 |
| | | | HM | 93.51 | $7.74 \times 10^6$ | −0.117 | 89.04 | 152.48 | 0.975 | 0.181 |

[a] = correlation coefficients of the Arrhenius plots and [b] = standard deviation.

*3.7. Antimicrobial Investigation*

The microbiological activity of **L** and its chelates was analyzed using the disc diffusion approach against pathogenic species of Gram-positive bacteria (*M. luteus* and *S. aureus*), Gram-negative bacteria (*E. coli* and *S. typhi*), and one strain of fungus (*C. albicans*). The results of the microbial screening for the synthesized compounds are shown in Table 7 and Figure 1. Various comparisons with the well-known standard **L** were made in order to complete the assessment of the biological activity of the synthesized chelates, and the results are presented in Table 7. Complexes (**2**) and (**3**) showed significant activity against two species of bacteria (*M. luteus*, *S. aureus*) and one strain of fungi (*C. albicans*). Studies have found that chelating compounds, when complexed with diverse metals, can impede the growth of bacteria. As a result, research into the antibacterial effects of transition metal complexes has been extensively conducted [65]. The resulting chelates were found to possess remarkable antibacterial capabilities. It is noteworthy that the biological activity of these compounds was enhanced upon chelation with metal ions. The core metal atom's polarity is diminished in chelated complexes due to the partial sharing of its positive charge with the ligands, and the entire chelated ring exhibits electron delocalization [66–69]. According to the chelation idea, chelation could actually make it easier for the complexes to pass through a phospholipid membrane. According to our findings, complexation improves the antibacterial activity [70,71]. The activity indexes for **L** and its chelates were graphically found from Figure 2.

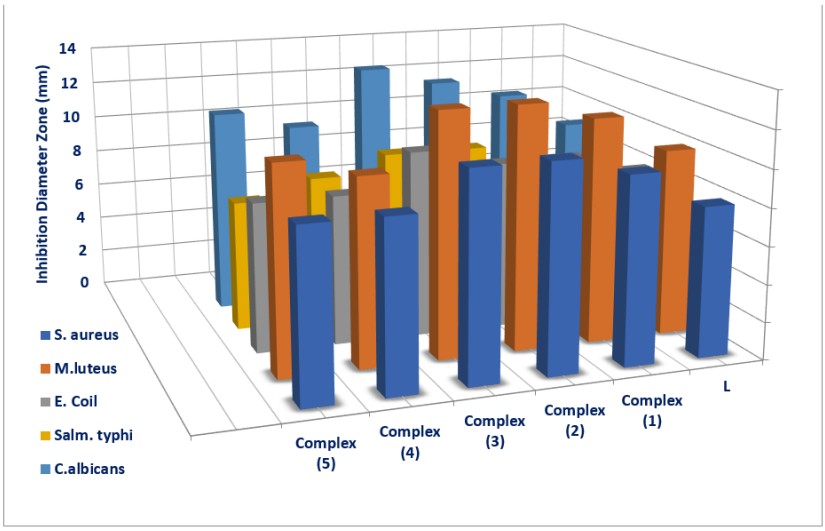

**Figure 1. L** and its chelates' biological activities.

**Table 7.** Activity index (%) and inhibition diameter zone (mm) measurements for **L** and its chelates.

| Tested Compounds | Examined Microorganisms | | | | | | | | | |
|---|---|---|---|---|---|---|---|---|---|---|
| | G(+ve) Bacteria | | | | G(-ve) Bacteria | | | | Fungi | |
| | S. aureus | | M. luteus | | E. coli | | S. typhi | | C. Albicans | |
| | D.Iz [a] (mm) | AI [b] (%) | D.Iz (mm) | AI (%) | D.Iz (mm) | AI (%) | D.Iz (mm) | AI (%) | D.Iz (mm) | AI (%) |
| L | 8 ±0.17 | 32 | 10 ±0.08 | 34.48 | 8 ±0.12 | 23.52 | 8 ±0.14 | 25 | 9 ±0.19 | 50 |
| (1) | $10^{NS}$ ±0.79 | 40 | $12^{NS}$ ±0.73 | 41.37 | $9^{NS}$ ±0.36 | 26.47 | 8 ±0.46 | 25 | $11^{NS}$ ±0.53 | 61.11 |
| (2) | $11^{+1}$ ±0.34 | 44 | $13^{NS}$ ±0.40 | 44.82 | $9^{NS}$ ±0.25 | 26.47 | $9^{NS}$ ±0.28 | 28.12 | $12^{+1}$ ±0.39 | 66.66 |
| (3) | $11^{+1}$ ±0.28 | 44 | $13^{+1}$ ±0.43 | 44.82 | $10^{NS}$ ±0.59 | 29.41 | $9^{NS}$ ±0.32 | 28.12 | $13^{+1}$ ±0.57 | 72.22 |
| (4) | $9^{NS}$ ±0.37 | 36 | 10 ±0.17 | 34.48 | 8 ±0.22 | 23.52 | 8 ±51 | 25 | $10^{NS}$ ±0.29 | 55.55 |
| (5) | $9^{NS}$ ±0.49 | 36 | $11^{NS}$ ±0.41 | 37.93 | 8 ±0.27 | 23.52 | 7 ±0.20 | 21.87 | $11^{NS}$ ±0.55 | 66.11 |
| Ciprofloxacin (control) | 25 ±0.3 | 100 | 29 ±0.2 | 100 | 34 ±1.11 | 100 | 32 ±0.98 | 100 | 0 | 0 |
| Nystatin (control) | 0 | 0 | 0 | 0 | 0 | 0 | 0 | 0 | 18 ±0.42 | 100 |

([a]) D.Iz (mm): inhibition diameter zone in millimeters; ([b]) AI (%): activity index for the examined compounds; $P^{+1}$ P significant and $P^{NS}$ P not significant.

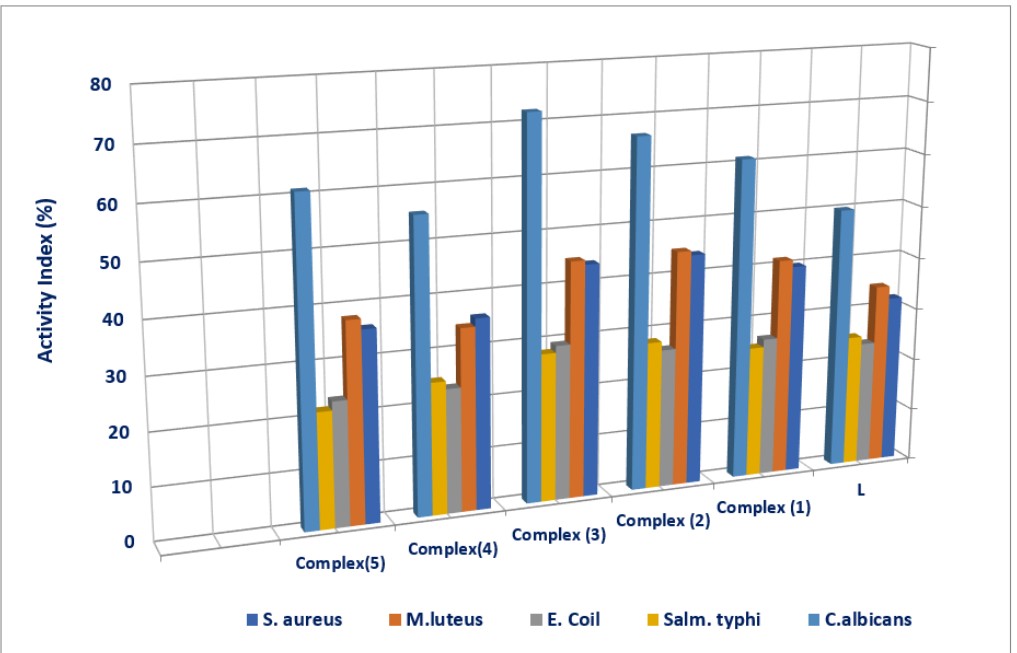

**Figure 2.** Activity index percent for **L** and its chelates.

## 4. Conclusions

Physicochemical in addition to spectroscopic methods were utilized to generate and characterize the five new metal chelates of **L** with lanthanum(III), zirconium(IV), yttrium(III), copper(II), and cobalt(II). According to the molar conductance, all complexes are electrolytic in nature. The **L** ligand interacted with metal ions as a bidentate in all the complexes via the oxygen atom of the phenolic group in addition to the nitrogen atom of the azomethine group according to the FT-IR spectra. Thermal analysis data indicate the existence of coordinated or uncoordinated water in the chelates. Thermodynamic parameters such as Ea, ΔH*, ΔG*, in addition to ΔS* have been performed via Coats–Redfern and Horowitz–Metzger techniques. Several bacterial and fungal species have been examined for **L**'s biological activity, along with its chelates. Yttrium(III) in addition to copper(II) chelates are effective as antibacterial agents compared with free **L** against two species of bacteria (*M. luteus*, *S. aureus*) and one strain of fungi (*C. albicans*).

**Supplementary Materials:** The following supporting information can be downloaded at: https://www.mdpi.com/article/10.3390/compounds3030029/s1. Figure S1: FT-IR spectra for **L** and its metal complexes; Figure S2: UV-Vis. spectra for **L** and its metal complexes ($1 \times 10^{-3}$ M in DMSO); Figure S3: 1H NMR spectra for **L** and its metal complexes ($1 \times 10^{-3}$ M in DMSO-d6); Figure S4: TG and DTG diagrams for **L** and its metal complexes; Figure S5: IR spectra for CuO and $ZrO_2$; Figure S6: Powder XRD pattern for $ZrO_2$; Figure S7: Kinetic parameters diagrams for **L** and its metal complexes.

**Author Contributions:** Conceptualization, S.M.A.E.-H. and S.A.S.; Formal analysis, A.A.N. and A.A.M.; Investigation, N.G.R.; Methodology, A.A.N. and A.A.M.; Supervision, S.M.A.E.-H. and S.A.S.; Validation, A.A.N. and S.A.S.; Visualization, A.A.M.; Writing—original draft, N.G.R., A.A.N. and A.A.M.; Writing—review and editing, S.A.S. All authors have read and agreed to the published version of the manuscript.

**Funding:** This research received no external funding.

**Institutional Review Board Statement:** Not applicable.

**Data Availability Statement:** We declare that our research does not involve any human subjects, human materials, human tissues, or human data.

**Conflicts of Interest:** The authors declare no conflict of interest.

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
