# Peer review of "First Report on Several NO-Donor Sets and Bidentate Schiff Base and Its Metal Complexes: Characterization and Antimicrobial Investigation"

_compounds, doi:10.3390/compounds3030029_

Round 1
Reviewer 1 Report
Comments and Suggestions for Authors
In this study, a Schiff base bidentate ligand (L) was synthesized through the condensation reaction of aniline and salicylaldehyde. L was successfully used to form complexes with lanthanum (III), zirconium (IV), yttrium (III), copper (II), and cobalt (II) metal ions. Physicochemical techniques including CHN analysis, Λ conductivity measurement, magnetic susceptibility investigations, FT-IR, and 1H NMR spectroscopy, UV-Vis. spectroscopy, and thermal studies (TG/DTG) were employed to characterize L and its metal chelates. The synthesized complexes showed promising antimicrobial activity, particularly the copper (II) and yttrium (III) chelates against S. aureus and M. luteus bacteria. The authors need to address some concerns before it is considered for published.
1. Section 3.4: The chemical shift of the proton in -N=CH- should be expected with a larger chemical shift rather than around 2.5 ppm. Please add peak integrations to Figure S3 and double check.
2. Section 3.5: “The thermal residues were differentiated by infrared spectra…and instead of the characteristic peaks for CuO and ZrO2 were found.”
Change “instead of” to “instead, ”.
3. Figure S4: Please add peak temperatures on DTG diagrams and weigh loss for each step on TG diagrams.
4. Figure S5 and S6: Add the reference IR spectra of standard CuO and ZrO2 as well as reference PXRD pattern for ZrO2. Add the PXRD pattern of CuO residue too, if possible.
Comments on the Quality of English Language
Minor edits are needed.
Author Response
Dear reviewer
Thanks so much for your revisions
General comments
The authors need to address some concerns before it is considered for published.
- Section 3.4: The chemical shift of the proton in -N=CH- should be expected with a larger chemical shift rather than around 2.5 ppm. Please add peak integrations to Figure S3 and double check.
Reply:
The signal observed at chemical shift around 8.95 ppm in all complexes assigned to -N=CH- and we made integration to this peak in Figure S3.
- Section 3.5: “The thermal residues were differentiated by infrared spectra and instead of the characteristic peaks for CuO and ZrO2 were found.”
Change “instead of” to “instead,
Reply:
The word “instead of” was corrected to “instead”.
- Figure S4: Please add peak temperatures on DTG diagrams and weight loss for each step on TG diagrams.
Reply:
We added peak temperatures on DTG diagrams and weight loss for each step on TG diagrams.
- Figure S5 and S6: Add the reference IR spectra of standard CuO and ZrO2 as well as reference PXRD pattern for ZrO2. Add the PXRD pattern of CuO residue too, if possible.
Reply:
We added the reference IR spectra of standard CuO and ZrO2 as well as reference PXRD pattern for ZrO2 with numbers 58, 59, 60. Unfortunately, the PXRD pattern of CuO residue is difficult to be done in addition to the yield of complex was finished.
Best regards
Sadeek A. Sadeek
The Corresponding Author

Reviewer 2 Report
Comments and Suggestions for Authors
The manuscript is well presented and clear enough. The characterization of the ligand and the complexes is complete. The manuscript can be published after the following minor revisions:
1) indicate the concentration of the UV and NMR spectra in the captions
2) report the UV spectra in terms of Molar Extinction Coefficient
3) report the assignments of the NMR signals also in the figures and not only in the tables
Author Response
Dear reviewer
Thanks so much for your revisions
General comments
The authors need to address some concerns before it is considered for published.
- Indicate the concentration of the UV and NMR spectra in the captions.
Reply:
We indicated the concentration of the UV and NMR spectra in the captions (Supplementary material file).
- Report the UV spectra in terms of Molar Extinction Coefficient.
Reply:
The molar extinction coefficient was reported in UV-Vis. spectra part (Page 7).
- Report the assignments of the NMR signals also in the figures and not only in the tables.
Reply:
The signals of 1H NMR was assigned in Figure S3.
Best regards
Sadeek A. Sadeek
The Corresponding Author
